# Field Efficacy of a Feed-Based Inactivated Vaccine against Vibriosis in Cage-Cultured Asian Seabass, *Lates calcarifer*, in Malaysia

**DOI:** 10.3390/vaccines11010009

**Published:** 2022-12-20

**Authors:** Zahaludin Amir-Danial, Mohd Zamri-Saad, Mohammad Noor Azmai Amal, Salleh Annas, Aslah Mohamad, Sutra Jumria, Tilusha Manchanayake, Ali Arbania, Md Yasin Ina-Salwany

**Affiliations:** 1Aquatic Animal Health and Therapeutics Laboratory, Institute of Bioscience, Universiti Putra Malaysia, Serdang 43400, Selangor, Malaysia; 2Department of Veterinary Laboratory Diagnosis, Faculty of Veterinary Medicine, Universiti Putra Malaysia, Serdang 43400, Selangor, Malaysia; 3Department of Biology, Faculty of Science, Universiti Putra Malaysia, Serdang 43400, Selangor, Malaysia; 4Department of Aquaculture, Faculty of Agriculture, Universiti Putra Malaysia, Serdang 43400, Selangor, Malaysia

**Keywords:** vibriosis, *Lates calcarifer*, feed-based vaccine, marine fish, field vaccination

## Abstract

**Simple Summary:**

Vibriosis is an important bacterial disease of marine fish that causes large mortality rates, resulting in substantial economic losses to the aquaculture industry. A new feed-based monovalent vaccine against vibriosis has been developed, but field efficacy of this vaccine has not yet been studied. This study determines the immune response and protective efficacy of the vaccine in Asian seabass that were kept in a field environment. The results revealed that the innate and adaptive immune responses were significantly higher in the vaccinated compared to the non-vaccinated groups.

**Abstract:**

*Vibrio* spp. are important aquaculture pathogens that cause vibriosis, affecting large numbers of marine fish species. This study determines the field efficacy of a feed-based inactivated vaccine against vibriosis in cage-cultured Asian seabass. A total of 4800 Asian seabass, kept in a field environment, were separated equally into two groups (vaccinated and non-vaccinated) in duplicate. Fish of Group 1 were orally administered the feed-based vaccine on weeks 0 (prime vaccination), 2 (booster), and 6 (second booster) at 4% body weight, while the non-vaccinated fish of Group 2 were fed with a commercial formulated pellet without the vaccine. Fish gut, mucus, and serum were collected, the length and weight of the fish were noted, while the mortality was recorded at 2-week intervals for a period of 16 weeks. The non-specific lysozyme activities were significantly (*p* < 0.05) higher in the fish of Group 1 than the non-vaccinated fish of Group 2. Similarly, the specific IgM antibody levels in serum and mucus were significantly (*p* < 0.05) higher in Group 1 than in Group 2, as seen in the second week, with the highest level 8 weeks after primary immunization. At week 16, the growth performance was significantly (*p* < 0.05) better in Group 1 and showed lower bacterial isolation in the gut than Group 2. Despite the statistical insignificance (*p* > 0.05), the survival rate was slightly higher in Group 1 (71.3%) than Group 2 (67.7%). This study revealed that feed-based vaccination improves growth performance, stimulates innate and adaptive immune responses, and increases protection of cultured Asian seabass, *L. calcarifer,* against vibriosis.

## 1. Introduction

Vibriosis is a disease caused by a group of Gram-negative bacteria of the genus *Vibrio,* and one of the most prevalent diseases in marine fish [1,2], leading to significant economic loss to the industry [3]. Fish infected with vibriosis exhibit a range of lesions, including eye lesions or blindness, gastro-enteritis, muscle necrosis, skin ulcers, and fin rot [4]. In Asia, particularly in the Southeast region, vibriosis is common and significant in farmed finfish and shellfish; some of the most notable ones are grouper (*Epinephelus* spp.), snapper (*Lutjanus* spp.), Asian seabass (*Lates calcarifer*), and shrimp (*Penaeus* spp.) [3,5,6,7,8].

In Malaysian aquaculture, the disease was first reported in Penang in the 1960s when high mortality rates were observed in brown-marbled groupers (*Epinephelus fuscoguttatus*) and giant groupers (*E. lanceolatus*) [9]. Major vibriosis outbreaks occurred in April and September 2010 on an Asian seabass farm in Sabah, Malaysia. The outbreaks caused high fish mortality, and *V. harveyi* was found to be the most abundant bacterium in the infected fish, water column, and biofilm [10]. Vibriosis was reported to occur mainly during the hatchery and grow-out phases [11], but adult fish may also be affected [12,13].

Traditionally, vibriosis is controlled using antibiotics [14]. However, with the growing concern about antimicrobial resistance, antibiotic usage is restricted in aquaculture, and vaccination is regarded as a way forward to control the disease [15]. This has led to considerable research activities to seek for efficacious vaccines to control vibriosis. Unfortunately, strains and antigenic variants of *Vibrio* spp. and the different serotypes have slowed the development of vaccines [16,17]. Moreover, in livestock, including fish, a vaccine must be designed to achieve three main components; the ability to confer immunity, ease of vaccine delivery, and widespread vaccine coverage [18].

Feed-based vaccination is regarded as a practical immunization method for fish farmers in aquaculture because it requires no special technical expertise to administer the vaccine and avoids direct contact between the handler and fish [19]. Additionally, feed-based vaccine delivery offers a less complicated, expensive, and stressful form of vaccination [20]. We have successfully developed a feed-based *V. harveyi* vaccine that provides cross-protection against *V. parahaemolyticus* and *V. alginolyticus* in laboratory trials [21,22]. However, the field efficacy of the newly developed vaccine needs further investigation.

Our earlier research on the oral inactivated *Vibrio harveyi* strain VH1 vaccine against vibriosis in hybrid grouper, *Epinephelus fuscoguttatus* × *E. lanceolatus* demonstrated that the vaccine could offer about 80% protection after field vaccination post-five days oral immunisation in an industrial farm in Langkawi, Kedah, Malaysia [22]. However, the feed-based inactivated vaccine has not been field tested in different hosts such as Asian seabass, *Lates calcarifer*, with a shorter period of vaccination and in a different geographical region to prove that the vaccine can confer broad protection in different hosts and locations. Its capacity to produce an antibody response in a different host, regimen, and setting was thus evaluated in the current investigation. Therefore, this work suggests the previously tested feed-based *Vibrio harveyi* strain VH1 vaccine with a new shorter vaccination regimen that might effectively protect *L. calcarifer* in the field against naturally occurring vibriosis, and might potentially enhance antibody response and growth performance of the cage-cultured marine fish.

## 2. Materials and Methods

### 2.1. Bacterial Strain and Growth Condition

Previously, the bacterial strain, *Vibrio harveyi* strain VH1 was isolated from tiger grouper, *Epinephelus fuscoguttus,* in deep sea-cages in Langkawi, Malaysia [23]. The strain was reported to induce strong antigenic responses to a homologous strain and cross-reacted with heterologous species of *V. parahaemolyticus* strain VPK1, *V. alginolyticus* strain VA2, and *Photobacterium damselae* strain PDS1 [23]. The *Vibrio harveyi* strain VH1 was initially cultured and maintained for 24 h at 30 °C on selective thiosulphate-citrate-bile-salts-sucrose (TCBS) agar (Oxoid, Hampshire, England) before the culture was further inoculated and incubated for 24 h at 30 °C with 150 rpm in tryptone soy broth (TSB) (Oxoid) + 1.5% NaCl.

### 2.2. Feed-Based Vaccine Preparation

A commercial feed pellet for marine fish (Star Feed, Star Feed Mills SDN. BHD., Klang, Malaysia; containing 43% protein) was ground with a blending machine to form fine mesh feed powder. Formalin-killed cells of *V. harveyi* strain VH1 were prepared following [22]. In brief, the bacterial suspensions were treated with 0.5% formalin and stored at 4 °C for 12 h or overnight. The inactivated bacterial suspensions underwent a 15-min centrifugation at 6000× *g* followed by three sterile phosphate buffer saline (PBS) washes. The bacterial pellets were then adjusted to a concentration of 10^8^ CFU/mL. Palm oil was added as an adjuvant to a final concentration of 10%. Finally, the vaccine mixture was thoroughly mixed and impregnated into the fish feed powder using an industrial mixer (Golden Bull-B10-A Universal Mixers, Johor Bahru, Malaysia), loaded into a mini feed pellet machine (Golden Avill, Guangdong Province, China) to obtain feed pellet with the size of 1 cm × 0.5 cm, and dried for 48 h at 30 °C. A patent application has been made for the oral vaccine’s composition and procedure (MyIPO Malaysia, patent No.: PI2021000105). In contrast, the non-vaccinated group was fed commercial feed containing PBS and 10% palm oil.

### 2.3. Ethics Approval

Ethics approval for this work was granted by the Institutional Animal Care and Use Committee, Universiti Putra Malaysia, under approval number: UPM/IACUC/AUP-R078/2019. All procedures in this study involving animals were performed following guidance of the Department of Biosafety, Ministry of Natural Resources and Environment, Malaysia.

### 2.4. Study Location and Study Design

A field-scale experiment was conducted in a private commercial fish farm located in the west region of Malaysia with reported vibriosis cases. A total of 4800 Asian seabass (*Lates calcarifer*) with an average body weight of 182 ± 31 g were selected. Prior to the start of the study, 15 Asian seabass were randomly dissected to detect the presence of bacteria, to ensure that the fish were healthy. The fish were found to be in good health, and no bacteria was found in any of the fish that were inspected.

The experimental design and vaccine regimen are shown in Figure 1. The experiment was conducted in duplicate by feeding the two cages with the prepared feed-based vaccine and the other two with non-treated control (PBS). At the start of the trial, the fish were fasted for 12 h prior to the vaccination to ensure maximum uptake of the feed-based vaccine. Then, the fish were fed with the respective feed for three consecutive days in weeks 0 (prime vaccination), 2 (first booster), and 6 (second booster) at the rate of 4% body weight. On other days, all fish were fed with the regular untreated commercial feed pellets (Star Feed, Klang, Malaysia) until the end of the 16-week experimental period.

Samples for lysozyme and antibody analyses (serum and mucus) were collected at 2-week intervals from 15 fish of each group to determine the antibody levels, while bacterial samples were isolated from the gut. In addition, water quality assessments were made using YSI Pro Plus (Yellow Spring Instrument, Yellow Spring, OH, USA) for pH, temperature, salinity, and dissolved oxygen, while ammonia nitrogen was assessed using a spectrophotometer (HACH Company, Loveland, CO, USA). Clinical signs and fish mortalities were logged before the rate of survival was determined after the 16-week study. Nevertheless, at 2-week intervals, the body weight of 10 randomly selected fish from each group was calculated to the nearest 0.1 g.

### 2.5. Growth Performance

Every two weeks for 16 weeks, the total body weights and length of experimental fish were measured from ten randomly sampled fish from each group. Growth response and feeding parameters, including total weight gain (WG), specific growth rate (SGR), feed conversion ratio (FCR), and feed efficiency (FE), were calculated using the formulas below [24]:WG=Final body weight (g)− Initial body weight (g)
$$\mathrm{SGR}\;\left(\%/\mathrm{day}\right)=100\times \left[\frac{\left(\ln\mathrm{Final}\;\mathrm{body}\;\mathrm{weight}\;\right)-\left(\ln\mathrm{Initial}\;\mathrm{body}\;\mathrm{weight}\right)}{\mathrm{Duration}}\right]$$
$$\mathrm{FCR}=\frac{\mathrm{Total}\;\mathrm{Feed}\;\mathrm{intake}\;\left(g\right)}{\mathrm{Weight}\;\mathrm{gain}\;\left(g\right)}$$
$$\mathrm{FE}=\frac{\mathrm{Weight}\;\mathrm{gain}\;\left(g\right)}{\mathrm{Total}\;\mathrm{Feed}\;\mathrm{intake}\;\left(g\right)}$$

### 2.6. Sample Processing

#### 2.6.1. Isolation and Identification of Bacteria

Gut samples were streaked onto thiosulphate-citrate-bile-salts-sucrose (TCBS) agar (Oxoid, Hampshire, UK) and were incubated at 30 °C for 24 h. Following incubation, *Vibrio* isolates, from their colonies that appeared rounded and medium-size yellow or green, were identified using morphological characterization and biochemical tests (API 20E, BioMerieux UK Ltd., Basingstoke, UK). The rate of bacterial isolation was calculated based on the percentage (%) of the number of vaccinated and non-vaccinated fish that were infected with *Vibrio* sp.

#### 2.6.2. Humoral Non-Specific Immune Parameters in Mucus and Serum Samples

The mucus and serum lysozyme levels were determined according to a previously described method [25]. Briefly, a total of 25 µL of the body mucus and serum was added into 75 µL of lysozyme-sensitive Gram-positive bacterium, *Micrococcus lysodiekticus* (Sigma-Aldrich, St Louis, MO, USA), prepared with 0.1 M phosphate citrate buffer and pH 6.3 in wells of a 96-well plate in triplicate. Following rapid mixing, the change in absorbance was measured at 450 nm after 30 s and 5 min using a Multiskan spectrum microplate reader (Multiskan™ GO Microplate Spectrophotometer, Thermo Fisher Scientific Inc., Madison, WI, USA). A unit of lysozyme activity was described as the amount of enzyme that caused a decrease in absorbance of 0.001 per minute and expressed as a U/mg unit.

#### 2.6.3. Detection of *V. harveyi*-Specific IgM in Serum and Mucus with ELISA

The enzyme-linked immunosorbent assay (ELISA) protocol for detecting *V. harveyi*-specific antibodies in serum and mucus in fish has been described previously by [26] and was used with minor modifications. Briefly, 96-well flat-bottom microtitre plates were coated with 100 µL coating antigens containing 10^5^ CFU/mL of *V. harveyi* strain VH1 in carbonate–bicarbonate buffer per well and kept at 4 °C overnight before being washed two times with sterile washing buffer (PBST consisted of PBS + 0.05% Tween 20). After washing, 200 L of blocking solution (PBS + 0.05% Tween 20 + 1% bovine serum albumin (BSA) (Sigma Aldrich, St. Louis, Missouri, USA) was added to each well to block non-specific binding. Each well was held for 1 h at 37 °C before being washed twice with PBST. The mucus and serum samples were then each diluted (1:1000) in PBST, and the resulting suspension (100 L) was added to three replicate wells of microtitre plates. These plates were then incubated once more at 37 °C for 1 h. Following three rounds of PBST washing, the treated plates were incubated for 1 h with anti-Asian seabass IgM monoclonal antibody (Aquatic Diagnostics Ltd., 1/33 in PBS) before being incubated for 1 h with anti-mouse-HRP (1/5000, Nordic). After the plates were washed thrice with PBST, 100 µL of TMB substrate solution (Thermo Fisher Scientific) was added to each well, and the reaction was held for 30 min at 37 °C. The reaction was stopped by adding 0.2 mol/L sulphuric acid, and the plates were analyzed immediately using a Multiskan spectrum microplate reader (Thermo Fisher Scientific) plate-reader at the absorbance at 450 nm.

### 2.7. Statistical Analysis

Data on bacterial isolation, lysozyme activity, antibody responses, and growth performance of the two groups were tabulated using Excel (Microsoft, Redmond, WA, USA). Results were achieved from five or more repeated samples and presented as mean ± standard deviation (SD). The data were then analyzed by one-way analysis of variance (ANOVA) and Tukey’s comparison of means. Statistical significance of differences were observed between the vaccinated and non-vaccinated groups and differences were considered statistically significant when *p* < 0.05 using IBM SPSS Statistics 26 (SPSS 26.0 package, SPSS Inc., Chicago, IL, USA).

## 3. Results

### 3.1. Growth Performance and Rate of Survival

By the start of the vaccination test between weeks 0 and 2, the average body length and body weight of fish in the two groups did not differ significantly (*p* > 0.05) (Figure 2). Following the first booster dose at week 2, the body length (Figure 2a) and body weight (Figure 2b) of the vaccinated Group 1 was significantly (*p* < 0.05) heavier compared to the non-vaccinated Group 2 from week 4 through to the completion of the 16-week research period. Similarly, Table 1 demonstrates that compared to the non-vaccinated Group 2, the vaccinated Group 1 had superior (*p* < 0.05) weight gain, specific growth rate, and feed efficiency with a lower feed conversion ratio. After the 16-week research phase, the vaccinated Group 1 recorded a higher survival rate of 71.3 ± 0.05% than the non-vaccinated Group 2 (67.7 ± 3.80%), although the difference was insignificant (*p* > 0.05).

### 3.2. Cumulative Mortality Rate

The cumulative mortality data were recorded from week 0 until week 16. Figure 3 shows the mortality patterns of the vaccinated and non-vaccinated groups throughout the vaccination trial. Between week 2 and week 16, mortality was greater in Group 2 (the non-vaccinated group) than in Group 1 (the vaccinated group). The mortality in both groups gradually increased but was insignificant (*p* > 0.05) between the two groups.

### 3.3. Isolation and Identification of Bacteria

As early as week 2 in both groups, *Vibrio* sp. was effectively recovered from the gut samples (Table 2). Throughout the research period, the non-vaccinated Group 2 had more intestinal bacterial isolates than the group that had received the vaccine, Group 1. The mean isolation of *Vibrio* spp. during the course of the trial was 22.50 ± 9.39% in Group 1, which was considerably (*p* < 0.05) lower than the non-vaccinated Group 2 at 46.67 ± 7.90%. The isolated bacteria were identified as *Vibrio alginolticus* (20%), *V. harveyi* (1%) and *V. coralii* (1%), while other isolated potential pathogenic bacteria included *Photobacterium damselae* and *Pasturella pneumotropica*.

### 3.4. Humoral Non-Specific Immune Parameters in Mucus and Serum Samples

The activity of lysozyme in the mucus increased significantly (*p* < 0.05) in the vaccinated group compared to the non-vaccinated group at weeks 4, 6, 8, and 10 (Figure 4a), while in serum (Figure 4b), the lysozyme activity in the vaccinated Group 1 was considerably (*p* < 0.05) higher than the non-vaccinated Group 2 at weeks 6, 8, and 16. The activity of lysozyme in the mucus of the vaccinated Group 1 peaked at week 8 (10.12 ± 3.18 U/mL) and in the serum at week 16 (22.59 ± 3.10 U/mL). Interestingly, at week 14, the serum lysozyme level in the non-vaccinated group was significantly (*p* < 0.05) higher than in the vaccinated group.

### 3.5. Detection of V. harveyi-Specific IgM in Serum and Mucus with ELISA

The mucus IgM levels against *V. harveyi* in all groups were insignificant (*p* > 0.05) before immunization (Figure 5a). Following respective immunization, the IgM antibody level of vaccinated Group 1 was significantly (*p* < 0.05) more progressive than that of the non-vaccinated Group 2 on week 2. However, the IgM antibody levels decreased from weeks 4 to 6 and were insignificantly different (*p* > 0.05) from the non-vaccinated Group 2. Following the second booster on week 6, the IgM levels increased significantly (*p* < 0.05) and peaked at week 8 before the IgM level started to decrease between weeks 10 and 14, but remained significantly (*p* < 0.05) higher than the non-vaccinated Group 2. On week 16, the IgM levels of both groups showed no significant (*p* > 0.05) difference.

The serum IgM response in vaccinated Group 1 displayed a similar pattern (Figure 5b). The serum IgM level in the vaccinated Group 1 was significantly (*p* < 0.05) higher as early as week 2 compared to the non-vaccinated Group 2. Similarly, the IgM level decreased at week 4 and was insignificant (*p* > 0.05) with the non-vaccinated Group 2 after the first booster dose administration. However, serum antibody levels increased after week 6 and peaked at week 8 before they gradually decreased thereafter but remained significantly (*p* < 0.05) higher than the non-vaccinated Group 2 until week 16.

## 4. Discussion

The Gram-negative bacteria *Vibrio* spp., which cause vibriosis in marine fish, are increasingly posing a serious danger to the aquaculture sector [27]. Vaccination, instead of drugs, has been demonstrated as a safe method of controlling infectious diseases [28]. However, due to the existence of several *Vibrio* species and strains, the development of a vaccine to prevent vibriosis has been delayed [16,29]. Monovalent vaccinations based on a single bacterial antigen are routinely utilised to treat diseases in fish. However, the immunisation technique for this type of vaccine necessitates the fish being immunised many times because some fish were infected with different diseases simultaneously and with more severity than a single infection [30]. Multiple fish vaccine handling causes stress to the fish and raises the cost of immunisation [31]. Given the cost and timing of fish vaccination, cross-protective vaccines that prevent multiple diseases in a single immunization are desirable [32,33,34].

Thus, in this study, we used a local isolate, *V. harveyi* strain VH1 from [23], which was previously isolated from tiger grouper, *Epinephelus fuscoguttus,* upon infection with vibriosis in deep-sea cages in Langkawi, Malaysia. The strain was reported to induce strong antigenic responses on its homologous OMPs antigen and cross-reacted with heterologous OMPs antigens from *V. parahaemolyticus* strain VPK1, *V. alginolyticus* strain VA2, and *Photobacterium damselae* strain PDS1 at a molecular weight of 32 kDa. The strain provided up to 83.3% protection against wild-type *V. harveyi* using *Artemia* sp. as a host model [23]. Refs. [35,36] employed the same strain in Asian seabass (*Lates calcarifer*) and brown-marbled grouper (*Epinephelus fuscoguttatus*), and found significant protection against *Vibrio* spp. Vaccination induces protection against specific infections by stimulating the immune system to limit and control the infection [37].

By associating it with protection, the antibody response is a frequently used metric to assess the effectiveness of fish immunization [38]. Additionally, a fundamental need for creating successful feed-based immunisation programs for Asian seabass is a knowledge of the adaptive immune responses in systemic and mucosal systems. In this study, stimulation of the immune systems was shown by the significant increase of IgM levels after immunization as early as week 2, it peaked at week 8, and remained significantly high until week 14. Therefore, the antibody response indicates that protection was possible until 14 weeks. The IgM tests of the immunised fish supported the idea that, in the field, vaccinations do not provide protection for the duration of the complete productive cycle [39], and in this study, it was around 14 weeks. The antibody pattern in the field was different compared to the experimental trial under a controlled environment [22]. Ref. [40] suggested that the differences in the antibody titre between laboratory conditions and the field trial may have resulted from fish cultured in different environmental backgrounds responding differently to vaccination. In this study, the vaccinated group showed a significant increment in IgM levels in their mucus and serum when vaccinated with the feed-based vaccine throughout the experimental period compared to the non-vaccinated group. This suggests the successful uptake and transport of antigen in the gut that involve both local and systemic immunities [19,41]. It was also documented that feed-based vaccination stimulates a high level of IgM antibodies in the mucus and serum [42]. Interestingly, after the first booster dose, the IgM antibody level was slightly decreased but higher than the non-vaccinated group. The results were similar to [43], who explained that the vaccine possibly could not stimulate enough memory cells to sufficiently boost up the secondary antibody following the administration of the booster dose.

Lysozyme activity has been commonly used as an immune biomarker [44]. Stress and diseases are some factors that influence the lysozyme activity in the host [45]. In fish, mucus and serum lysozyme are produced by leukocytes, and this enzyme plays a vital role in innate immunity because it lyses the cell wall of Gram-negative and Gram-positive bacteria [46]. This study recorded high lysozyme activity levels in the vaccinated group. Previous research by [47,48] also demonstrated high lysozyme stimulation post-oral vaccination. The authors stated that the non-specific immune responses via the lysozyme activity in the vaccinated group could positively affect the release of antigens in the host, thus stimulating the host immune cells in orchestrating a combined defence mechanism against bacterial challenge.

Similarly, Ref. [49] reported on the feed-based bivalent vaccine against *Streptococcus iniae* and *Aeromonas hydrophila,* which showed higher serum lysozyme activity in the vaccinated fish, with a range of 158–325 U/mL, compared to the unvaccinated control, with a range of 93–95 U/mL, from day week 0 to week 7. Furthermore, this study was similar to some studies on probiotics that reported high lysozyme activity in the mucus post-feeding with *Bacillus subtilis* and *Lactococcus lactis* in which the bacteria could stimulate a high innate immune parameter in fish following feeding with the feed containing the bacteria [50,51]. Lysozyme is one of the crucial innate immune factors because it helps fish become more resistant to diseases by increasing the number of phagocytes that produce it and the amount of it that is generated [52]. The result suggested that the innate system of vaccinated Asian seabass was induced and may provide a certain degree of disease resistance. In this current study, the result showed a slightly higher survival rate for vaccinated fish at 71.30 ± 0.05%, while survival in the non-vaccinated group was 67.70 ± 3.80%. The results may be due to no outbreak occurring during the vaccination period; thus, higher survival was seen in the non-vaccinated fish. Better survival after vaccination is helpful to farmers as it leads to a larger harvest. Further studies under laboratory challenge need to be conducted to determine the protection against the pathogenic *Vibrio* spp.

Bacterial isolations showed a significant reduction in the course of the 16-week research period. The overall incidence rate of *Vibrio* spp. throughout the study period, the vaccinated group was found to be lower than the control group. This was in agreement with previous studies, which recorded lower rates of pathogen isolation among vaccinated fish compared to those not vaccinated [19,37,53]. In this study, *Vibrio* spp. was isolated from the fish’s gut. Due to the aquaculture system’s open nature and water being the principal means of transmission of *Vibrio* spp. in fish, the bacteria may gain entry into the hose via the skin, gills, or gut [54,55]. As combating diseases and defending against infections requires a physiological cost, an “immune” host benefits from a decreased intensity of virulence of the bacteria as the fish may save more energy from carrying and harbouring pathogens, leaving more resources available for normal development. Therefore, vaccination that may reduce the abundance of bacteria can potentially encourage growth by dropping the metabolic load of the immune response to infection [56]. In addition, the altered gut microbiota of the fish brought on by the consumption of vaccine-tainted feed may increase nutrient absorption in the intestine and positively affect metabolism in addition to the healthy state of the immunised fish and the heightened immunity brought on by the vaccination [57]. This can be seen in better feed conversion efficiency and weight gain in the vaccinated group in this current study. The growth performance of the vaccinated Asian seabass was higher than that of the non-vaccinated fish, as shown by other studies [58,59]. The authors reported a significant enhancement in growth performance after feed-based vaccination.

The main factor determining the success of vaccine testing in the field is the good stimulation of the immune system in the fish at the time of vaccination [60]. However, other contributing factors, such as stocking density, natural environment, water quality, and the farm management system, could influence the development of a robust immune system without experiencing high-stress factors [61]. Thus, to improve vaccine efficacy, the contributing stress factors should be minimised [62].

## 5. Conclusions

Field application of the feed-based vaccine stimulates the specific and non-specific immune responses as well as improves the growth performance and survival of cultured Asian seabass, *L. calcarifer*. The present results suggest that the feed-based fish vaccine is potentially an effective vaccine candidate against vibriosis, especially suitable for small-scale private farmers who cannot afford labour-intensive vaccine protocols requiring special equipment. In addition, vaccination provides higher economic benefits, giving the farmers higher profits than the unvaccinated cultures.

## Figures and Tables

**Figure 1 vaccines-11-00009-f001:**
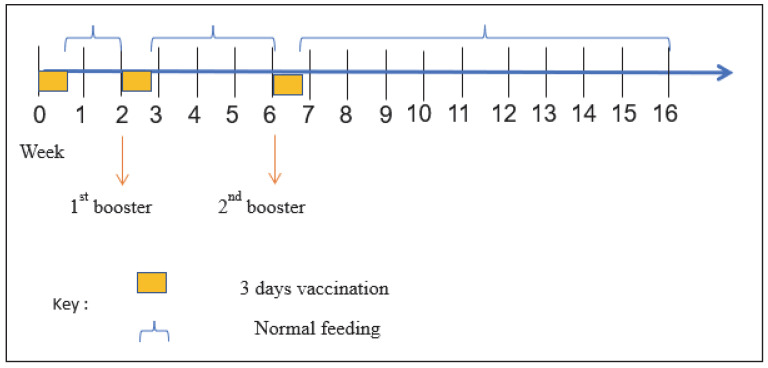
Experimental design and vaccination regimen using the feed-based vaccine against vibriosis.

**Figure 2 vaccines-11-00009-f002:**
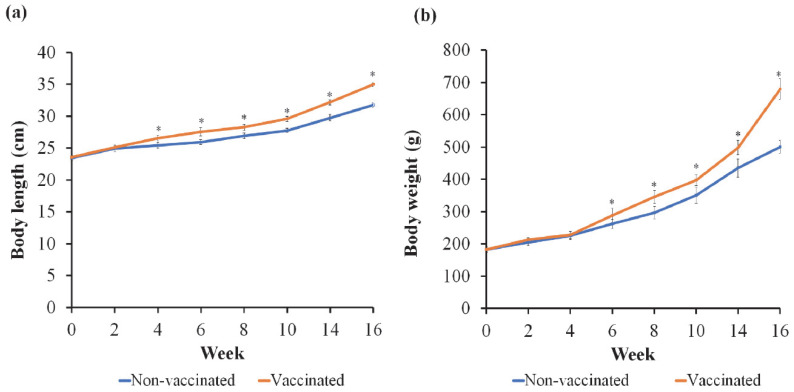
The body length (**a**) and body weight (**b**) of Asian seabass throughout the 16-week vaccine trial. Asterisks (*) indicate significant (*p* < 0.05) differences between the vaccinated and non-vaccinated groups.

**Figure 3 vaccines-11-00009-f003:**
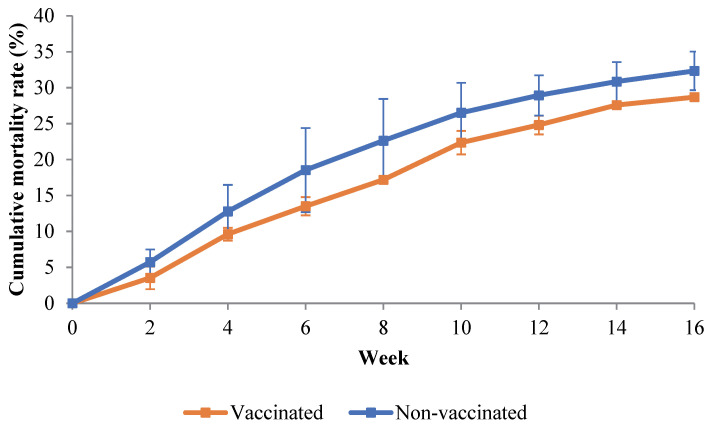
Cumulative mortality rate (%) of the vaccinated and non-vaccinated Asian seabass throughout the vaccination trial. Over the course of the trial, no change was found to be significant (*p* > 0.05).

**Figure 4 vaccines-11-00009-f004:**
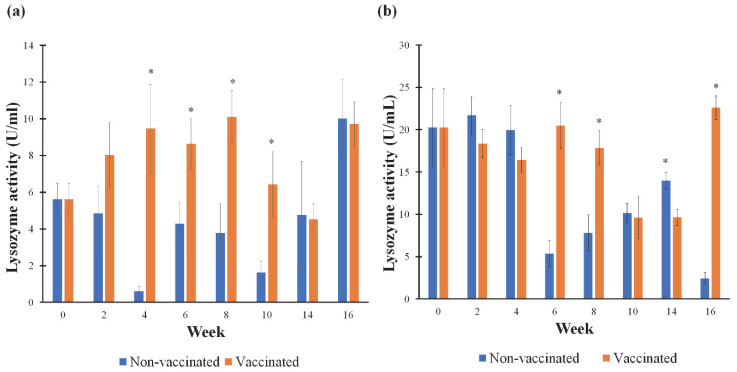
The activity of lysozyme in the mucus (**a**) and serum (**b**) following vaccination with feed-based inactivated *V. harveyi* strain VH1 vaccine. The non-vaccinated fish were given a commercial pellet formulated with PBS. Asterisks (*) indicate a significant difference at *p <* 0.05 between the vaccinated and non-vaccinated groups.

**Figure 5 vaccines-11-00009-f005:**
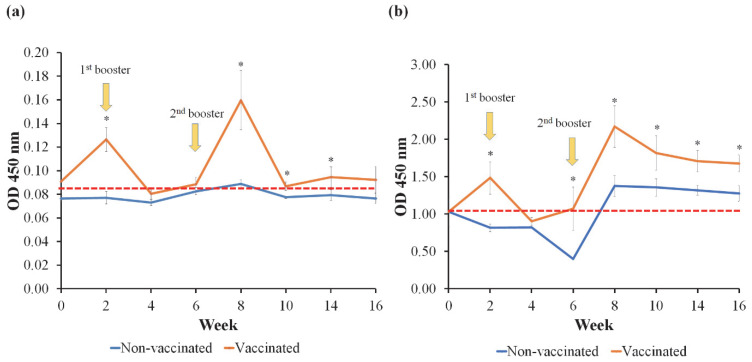
The mucus (**a**) and serum (**b**) IgM responses following vaccination with feed-based inactivated vaccine against vibriosis. The non-vaccinated fish were fed a commercial pellet formulated with PBS. Asterisks (*) indicate significant differences at *p* < 0.05 between the vaccinated and non-vaccinated groups.

**Table 1 vaccines-11-00009-t001:** Growth performance parameters and survival rate of Asian seabass fed with vaccinated feed for 16 weeks. Asterisks (*) indicate a significant difference (*p* < 0.05) from the non-vaccinated.

Parameters	Treatment Group	
	Non-Vaccinated	Vaccinated
Initial body weight (g)	182.00 ± 32.09	182.40 ± 30.15
Final body weight (g)	500.67 ± 79.12	680.00 ± 126.54 *
Survival rate (%)	67.7 ± 3.80	71.3 ± 0.05
Weight gain (g)	315.76 ± 34.79	498.94 ± 32.24 *
Specific growth rate (% day)	0.90 ± 0.02	1.18 ± 0.05 *
Feed conversion ratio	4.02 ± 0.44	2.78 ± 0.18 *
Feed efficiency	0.25 ± 0.03	0.36 ± 0.02 *

**Table 2 vaccines-11-00009-t002:** Rate of *Vibrio* spp. isolation (%) in the gut of Asian seabass. Asterisks (*) indicate a significant difference at *p <* 0.05 between the vaccinated and non-vaccinated groups.

Week	Gut	
	Non-Vaccinated (%)	Vaccinated (%)
0	0.00	0.00
2	46.67	20.00
4	53.33	26.67
6	46.67	26.67
8	46.67	26.67
10	40.00	26.67
14	60.00	26.67
16	46.67	26.67
Mean ± SD	46.67 ± 7.91 *	22.50 ± 9.39

## Data Availability

Not applicable.

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
