# Peer review of "Field Efficacy of a Feed-Based Inactivated Vaccine against Vibriosis in Cage-Cultured Asian Seabass, Lates calcarifer, in Malaysia"

_vaccines, 2022, doi:10.3390/vaccines11010009_

Round 1

Reviewer 1 Report

Dear authors,

Thank you very much for your interesting work. The manuscript has discussed the effect of a feed-based monovalent vaccine against Vibrio spp. for improving growth performance. The manuscript is well-written and has data. Therefore, I recommended considering acceptance after minor corrections.

1. "Introduction" mainly discusses Malaysia's aquaculture, however, the title is Asian seabass, with much bigger images. Therefore, the authors are suggested to change the title to suitable the work or add some more introduction concerning other ASEAN aquaculture.

 2. Have the authors considers the effects of the feed-based vaccine on the quality of the meat of fish, also how about the resident vaccine in the aquaculture environment and fish-farm wastewater treatment?

3. In the Discussion, the authors said there are various Vibrio species and strains it is hard to develop a specific vaccine. Therefore, I think it would be better to add some more discussion about the strain as well as a mixed vaccine to improve the treatment where is applicable.

4.  How many times repeat all the experiments? Please add this information to this manuscript which helps to increase the trust of data. 

5. Please add some footnotes for the table to easier understand the table (e.g. Table 2 (*) symbol)

6. There are a number of typos, please check and correct these typos to avoid these mistakes before publication (e.g. there is a space between number and unit,...)

Author Response

Dear Reviewer,

As attached.

Thank you.

Reviewer 2 Report

Dear authors 

The manuscript entitled "Field efficacy of a feed-based inactived vaccine against vibriosis in cage-cultured Asian seabass, Lates calcarifer" is suitable for publication in "Vaccines" after minor revision.

Some specific comments:

Materials and Methods section: 

- Know the authors if the PDS1 Phtotobacterium damselae strain is the subspcies piscicida or damselae?

- Know the authors the degree of virulence of the V. harveyi strain VHI?

Results section:

- Do you know the percentage of V. harveyi  in relation to the other bacterias isolated from the gut?. I guest it´s much smaller since the inactived vaccines is formulated against this pathogen. This should be indicated in the text.

- The data about the humoral non-specific immune parameters in mucus and serum sample should be more discussed, mainly the data corresponding to the 16 week.

Author Response

Dear Reviewer,

As attached.

Thank you.
